

# A method for creating interactive, user-resembling avatars

Igor Macedo Silva[1,2] and Renan C. Moioli[2]

[1] Centro de Tecnologia, Universidade Federal do Rio Grande do Norte, Natal, Rio Grande do Norte, Brazil
[2] Graduate Program in Neuroengineering, Edmond and Lily Safra International Institute of Neuroscience, Santos Dumont Institute, Macaiba, Rio Grande do Norte, Brazil

## ABSTRACT

Virtual reality (VR) applications have disseminated throughout several fields, with a special quest for immersion. The avatar is one of the key constituents of immersive applications, and avatar resemblance can provoke diverse emotional responses from the user. Yet a lot a virtual reality systems struggle to implement real life-like avatars. In this work, we propose a novel method for creating interactive, user-resembling avatars using available commercial hardware and software. Avatar visualization is possible with a point-cloud or a contiguous polygon surface, and avatar interactions with the virtual scenario happens through a body joint-approximation for contact. In addition, the implementation could be easily extended to other systems and its modular architecture admits improvement both on visualization and physical interactions. The code is under Apache License 2.0 and is freely available as Supplemental Information 1 in this article.

## INTRODUCTION

Virtual reality (VR) is understood as an advanced human–computer user interface that mimics a realistic environment by linking human perception systems with a virtual environment (*Zheng, Chan & Gibson, 1998*). The key point about VR is its attempt to provide seamless interaction with a computational environment, thus understanding human intent and creating a reasonable response from the virtual environment.

The gaming industry has historically driven developments on VR (*Mills, 1996*), contributing to lowering the cost of technology. There has been a growing demand for techniques or devices that improve the user's ability to feel inside a game, surrounded by elements that promote the desired gameplay. To achieve that, the industry has turned to motion tracking devices and immersive 3D graphics, including head mounted displays (HMDs), such as Oculus Rift, HTC Vive, and Hololens, and motion tracking technologies, such as Kinect, Wii Remote, Leap Motion, and Virtuix Omni VR treadmill.

This recent surge of affordable VR devices expanded the number of VR applications. To mention but a few examples beyond the gaming scene, VR is now widely used in medical applications for physical rehabilitation (*Paolini et al., 2013*; *Baldominos, Saez & Del Pozo, 2015*; *Draganov & Boumbarov, 2015*; *Morel et al., 2015*; *Donati et al., 2016*; *Shokur et al., 2016*), exposure therapy for phobias and post-traumatic stress (*Notzon et al., 2015*;

Corresponding author
Renan C. Moioli, moioli@isd.org.br

*Cuperus et al., 2016*), treatment for addiction (*Park et al., 2016*) and even autism (*Didehbani et al., 2016*). Other fields such as architecture and urban planning (*Portman, Natapov & Fisher-Gewirtzman, 2015*; *Luigi et al., 2015*) and education (*Abulrub, Attridge & Williams, 2011*) also benefit from the technology.

However, as advanced as those technologies are, there are a few limitations to each of them. The technologies mentioned above, such as the HMDs and motion tracking devices, tackle a single piece of the virtual reality interaction problem. The HMDs give visual support for the simulation while motion tracking devices provide means for our body to interact with the virtual world. On one hand, most HMDs deal exclusively with visual perceptions and head movement while completely ignoring any other body movement. As a result, HMDs applications are usually static and display static, generic avatars, frustrating any kind of user interaction other than through vision and head movement. Motion tracking devices, on the other hand, allow for whole-body user interaction but limit the immersion experience because they do not take into account the user's visual field. Therefore, users are limited on how they can interact with the system, depending on which device they use.

A possible solution is to use the capabilities of both devices in an integrated hardware-software approach. Ideally, this system would be able to track body movements, record user images and present them inside the head mounted display, reacting to what the user is looking at and how the user is moving.

A key issue is that there is scarce information on how to integrate and utilize both technologies in one whole system. An example is shown by *Woodard & Sukittanon (2015)*, explaining how both Oculus Rift and Kinect could be used to create virtual interactive walkthroughs for building projects. However, they do not use the Kinect's full capabilities, such as body-frame detection and RGBA frames to recreate the avatar and interact with the virtual environment.

In accordance with growing evidence of the importance of avatar representation and interaction to obtain an improved immersion experience in virtual reality, this work presents a software approach to integrate a motion tracking device with depth sensing capabilities (Kinect V2) to a head mounted display (Oculus Rift DK2) in order to improve the interaction and immersion perception in VR systems. This software uses the SDKs of both devices, along with OpenGL for rendering graphics and Bullet Physics for physic simulations, to create a highly resembling 3D representation of the user in the virtual space. This 3D representation, henceforth referred to as avatar, is highly sensitive to the user's movements and can interact with virtual objects in the virtual environment. The main result is a demo project that incorporates all of the necessary code to compile a neutral virtual environment with objects and the avatar. The code is under Apache License 2.0 and is freely available as supplementary material. We believe that this work contributes to easily integrate HMDs and motion tracking devices to expand real time immersive experience interaction.

# BACKGROUND AND RELATED WORK

## Virtual reality technologies

Among the devices used on VR research, the Oculus Rift and the Kinect are probably the most widely known products. Both devices tackle specific problems underlying human–computer interfaces and, by the time of their launch, they brought innovative solutions to the common consumer.

Nowadays, there are a few other solutions that aim at offering the same range of features and usability of Oculus Rift and Kinect. Regarding depth cameras, such as the Kinect, we can cite the Asus Xtion, DepthSense 525 Camera, and LeapMotion as alternatives. Although LeapMotion is the most accurate in comparison to the others, with spatial resolution of about 1 mm (*Weichert et al., 2013*), it has the shortest range of action of 0.6 m (*LeapMotion, 2017*). Another device called the DepthSense 525 Camera is able to provide 60 fps and depth ranging from 15 cm to 1 m, but its frames per second throughput diminishes quickly as the depth range increases, going from 2.5 m at 15 fps to 4 m at 6 fps (*DepthSense, 2017*). The Asus Xtion has a similar precision to the Kinect V1, as shown in *Gonzalez-Jorge et al. (2013)*, and its most recent version is comparable to the Kinect V2 in terms of technical specifications (*ASUS, 2017*; *Microsoft, 2017b*). Yet the Kinect has a much broader community and support from Microsoft for its official SDK. This reflects specially in the number of applications and publications made with Kinect or Asus Xtion. Thus, considering the specifications shown here, we choose to use in our system the Kinect V2 as the main interface for depth sensing.

The first to appear on the market was the Kinect v1, in 2010, with the official software development kit (SDK) released later on in 2011 (*Microsoft, 2017a*). In 2014, a second iteration of the Kinect was released (*Microsoft, 2017b*). The Kinect V2 has a 1,080 p color camera operating at 30 Hz in good light conditions, a $512 \times 424$ depth sensing camera operating at 30 Hz and $70 \times 60$ field of view, sensing from 0.5 to 4.5 m, and active infrared capabilities with the same resolution.

With its pose tracking capabilities and affordability, the Kinect quickly made its way to the research scene. Regarding accuracy for motion tracking, the Kinect is adequate for most scientific purposes, including medical studies on posture and rehabilitation (*Clark et al., 2012*; *Clark et al., 2015*; *Zhao et al., 2014*). In particular, *Otte et al. (2016)* have shown that the Kinect V2 can derive clinical motion parameters with comparable performance to that obtained by a gold standard motion capture system.

As for virtual reality headset, the Oculus Rift was for a period the only option available. But, soon enough, companies started to unveil similar products and nowadays the Oculus Rift has a few competitors. Some of them have yet to be commercially launched, such as the Fove and Star VR. Two commercially available platforms are the Playstation VR and the HTC Vive. Both have very similar hardware specifications to the Oculus Rift (*Playstation, 2017*; *DigitalTrends, 2017*). The main difference is the price range: while the HTC Vive is priced at U\$ 799, the Playstation VR costs U\$ 386 and the Oculus Rift is sold for U\$ 499. Yet, the Oculus Rift has been in development since 2012 and has created a solid developer

community. For this reason, the Oculus Rift is a sensible choice both in terms of hardware specifications and development support.

The first release of the Oculus Rift, named DK1, was for supporters only and occurred in 2012. The second iteration was released in 2014 as DK2, and the first commercial version was released in 2016. The Oculus Rift DK2 improved the first iteration with a screen resolution of 960 × 1,080 p per eye, a refresh rate of 75 Hz and persistence of about 2 to 3 ms. In addition, the head-mounted display has sensors to detect head motion both internally and externally.

## The immersion requirement

If we understand VR as a human–computer interface, there are two core ideas that measure the quality of the interface: immersion and interactivity (*Zheng, Chan & Gibson, 1998*). Immersion occurs when the user is able to block out distractions, usually any perceptions from the real world, and then focus on the virtual elements with which one wants to work. Interactivity is the possibility to interact with the events of the virtual world. Both ideas are connected and interdependent. For example, if a VR system has low interactivity, the immersion factor might be affected. On the other hand, if the system is not immersive, the interactivity will be less engaging.

The immersion requirement dictates the overall experience with a VR system. A previous study developed by *Alshaer, Regenbrecht & O'Hare (2017)* identified three immersion factors which affect perception and the sense of self-presence in a VR system: the display type (head mounted display (HMD) or monitor), field of view (FOV) and the user's avatar. A poor tuning of these factors can cause discomfort and a sense that the virtual actions or events are absurd and do not reflect reality among users, eventually disrupting the immersive experience and causing a poor VR experience as a whole.

## Avatar embodiment

In VR systems, the avatar consists of a visual representation of the user, a construction made to visually communicate aspects of one's identity and how he or she wants to be perceived. Using the avatar, the user is able to give meaning to their virtual self and that correlates with the sense of presence (*De França & Soares, 2015*). The concept of presence, as mentioned by *De França & Soares (2015)*, allows a discussion about self representations at three different, yet interdependent levels: body, affection and identity. Focusing on those three levels is specially important in VR applications in order to improve the user's performance in the system. In the discussion section, we present how the developed system exploits those levels in order to create sense of embodiment.

This connection between avatar and physical self has been investigated. *Wrzesien et al. (2015)* focused on how teenagers can learn to regulate their emotions by watching physically similar avatars deal with frustration. The authors concluded that when the avatar is modeled to resemble the human user, the intensity of the emotional states, emotional valence and arousal for each participant has a much greater level than that obtained when the avatar is neutral. This confirms that the similarity between the avatar and the user may cause a significant psychological response.

Another study (*Fox & Bailenson, 2009*) was concerned about how this similarity might modify behavior. After dividing the participants in groups with a virtual self representation or with another person representation, they were exposed to voluntary sessions of exercises with rewards and punishments. The research concluded that the group with virtual self representation exercised significantly more regardless of reward or punishment. Both studies demonstrate that the avatar influences the overall experience on VR and expands this experience to the user in the physical world. Similar results were obtained in a study by *Lugrin, Landeck & Latoschik (2015)*, confirming that behavior and performance are indeed altered by avatar representation.

## Applications

The avatar is a core concept for immersion in VR (*Alshaer, Regenbrecht & O'Hare, 2017*), thus most VR applications benefit from improved user-resemblance and interactivity with the virtual environment.

In the medical field, VR is broadly used for rehabilitation treatments. For example, the systems presented by *Rudinskiy et al. (2014)*, *Donati et al. (2016)* or *Shokur et al. (2016)* use VR with an avatar visualization for gait rehabilitation. In this case, an improved avatar visualization would help generating a better sense of immersion and could possibly help with the treatment. This improvement is thought to come from an increased engagement in the rehabilitation activities, which provokes intense and life-like emotions and behaviors. This motivates the user to perform the activities.

Another field that benefits from this technology is social studies. In a controlled environment, such as a virtual environment, it is possible to reproduce a variety of social phenomenons. A highly user-resembling avatar may improve the immersion experience and mimic social interactions. For example, *Stavroulia et al. (2016)* creates a simulated environment to reproduce bullying occurrences inside a classroom. Considering the benefits of an avatar representation, it is expected that such a simulation would benefit from the avatar generation method described here by improving on the overall experience of the system.

As a last example, tourism or even architecture could benefit from this technology by giving the user an immersive experience in which they could feel immersed. It is probably easier to create a sense of space and dimensions if you can refer to your own body as a reference. In this sense, interior design professionals and students could use our method to showcase a project and literally place the observer inside the space to visually comprehend each detail of the project. In tourism, virtual environments can replicate interesting places and allow the user to see and even interact with elements through its own avatar.

# MATERIALS & METHODS

## Hardware and development environment preparation

Hardware integration from different manufacturers is a key point in VR applications and may increase software development complexity.

The Oculus Rift DK2 was chosen for visualization and the Kinect for Windows V2 was chosen for motion tracking, color and depth sensing. This choice considered that both

**Table 1  Compatibility requirements for development environment and software usage.** System relevant information: Windows 10 Pro; AMD Radeon HD 7800 Series; Intel(R) Core(TM) i7-3770 CPU @ 3.40 GHz; 24 GB RAM.

| Identification | Compatibility requirement description |
| --- | --- |
| C1 | The Kinect for Windows V2 requires Windows 8 or higher. |
| C2 | The Kinect for Windows V2 requires Kinect for Windows SDK 2.0. |
| C3 | The Oculus Rift DK2 on Windows 10 requires Oculus Runtime and SDK 0.7 or higher. |
| C4 | The Oculus Runtime 0.8 running alongside an AMD Radeon HD 7800 requires AMD Catalyst 15.11.1 beta driver[*] |

Notes.

[*]Later versions of the new Crimson Driver were found to be incompatible with Oculus Runtime 0.8.

devices were the latest iteration available at their respective product lines by the time this study was developed and, thus, had significant improvements over previous iterations. Specifically, the Oculus Rift DK2 has a higher screen resolution and better head movement tracking than the DK1, which contributes to an improved and seamless visual interaction with the virtual environment.

This choice of hardware requires a specific setup of the development environment and later usage of the software because there are potential compatibility issues with operational system, hardware runtime routines, graphic card drivers and available SDKs for each device. A list of requirements for compatibility is available in Table 1 along with relevant information from the system. When multiple versions of software were possible, we opted for the most recent.

The language of development is C++ for it was the only language officially supported by both Oculus Rift and Kinect SDKs. Both SDKs can be obtained through their respective product websites, along with correct versions of drivers and runtimes (shown in Table 1) which are necessary for hardware functioning. In addition to SDKs, drivers and runtimes, we used a few libraries: the OpenGL library is already integrated to the Oculus SDK and sample programs; the Bullet Engine 2.83.7 is responsible for physics simulation in the virtual environment, specially collisions between the avatar and virtual objects; and the OpenMP 2.0[1] is used for parallelization in the handle classes for each SDK. The code was implemented and compiled with Visual Studio 2013.

## Software requirements

The requirements listed in Table 2 guided software development. Other functions (only feasible for the chosen hardware) were implemented along the development in order to provide more interaction options to the user and will be described later.

Regarding requirement R1.1, the decision to create the avatar based on real time data is crucial to the final objective of the software, which is to create a deeper sense of immersion in the user. This requirement should enforce the avatar to morph and move accordingly to each user in real time and provide plausible responses to each of their actions in the virtual environment.

[1]This is the highest OpenMP version available for Visual C++ in Visual Studio 2013.

**Table 2  Basic Requirements for the development of the software.**

| Identification | Requirement description |
| --- | --- |
| R1 | The software *shall* create a 3D recreation of the user—known as the avatar. |
| R1.1 | The avatar *shall* be defined by real time data from the depth and color sensors. |
| R1.2 | The avatar *shall* be displayed in a 3D virtual environment in a HMD. |
| R2 | The software *shall* offer the option to visualize in first-person perspective or third-person perspective. |
| R3 | The software *shall* render at an acceptable frame rate (minimum 24 fps). |
| R4 | The software *shall* provide simulated physical interaction between the avatar and the environment. |
| R5 | The software *shall* provide a keyboard interface to select which user has the Oculus Rift point of view, when first-person perspective is selected. |
| R6 | The software *shall* provide a keyboard interface to adjust the camera position in order to accommodate and match the user virtual visualization with the real world. |

## Implementation of core functionality

The development of the main idea of the software, described by requirement R1.1 in Table 2, involves the concept of a point cloud (*Linsen, 2001*; *Rusu & Cousins, 2011*). This element is composed by a group of points fully defined by their location in a 3D space with magnitude on each of the three space dimensions. Such property allows the programmer to plot this point cloud in a 3D virtual environment and visualize the surface from which the point cloud was acquired in the first place. When visualizing the point cloud, it is possible to perceive the dots as a "cloud" floating in the virtual space, hence the name. If not computationally generated, one can acquire a point cloud from the real world with a depth sensor, laser scanner data or photogrammetric image measurements (*Remondino, 2003*). For this purpose, this paper uses the Kinect V2, which provides features beyond depth sensing that will be discussed later.

The rationale is that it is possible to create a polygon surface from the point cloud of one user. Therefore, the first step was to acquire the point cloud and plot it in the virtual environment, with the aid of the OpenGL library. Using the Kinect SDK, we developed a handle class to request data from the Kinect and translate it to an OpenGL compatible format. The Kinect V2 can simultaneously detect and differentiate up to six users, also called bodies. In addition, the Kinect provides five types of pertinent data for this project:

1. Color Frame: a RGB frame with 1,920 × 1,080 p.
2. Depth Frame: a depth frame with 512 × 424 p.
3. Body Frame: a 512 × 424 frame where each element contains the index of the detected body for the relative pixel in the depth frame.
4. Joint Vertices: a float array containing information for the camera space[2] localization of each joint of each body.
5. Body Array: indicates which of the six possible bodies the Kinect is currently tracking

The acquisition of each type of data is executed by a function inside the Kinect Handler class and parallelized with the OpenMP library for optimization. Figure 1 shows the function GetColorDepthAndBody() called from the handle class, which returns the frame

[2]The camera space is a three-dimensional space that represents the position of each point in the real world "space".

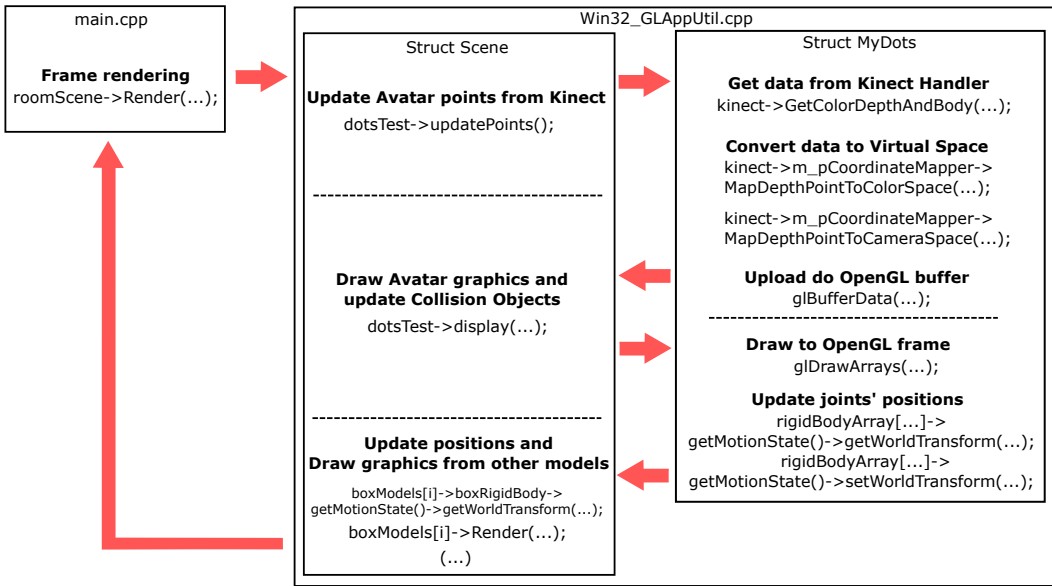

**Figure 1** Frame Rendering Cycle: structures and functions.

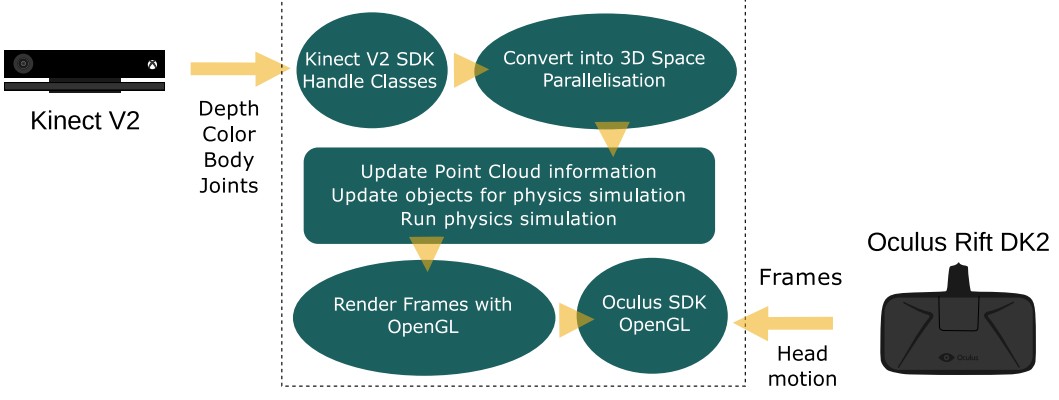

**Figure 2** Dataflow and processing.

data from the kinect. In Fig. 2, it it possible to observe this same process with a higher level of abstraction, where the handle class acquires the kinect frame data and returns it to the next structure for processing.

Next, the depth information from the depth frame is mapped to camera space in order to create the aforementioned point cloud, as shown in Fig. 2. This means that each pixel in the depth frame is mapped to a 3D virtual representation of the real world, so that the software is able to plot a visualization for the point cloud. Alongside, it is also needed to map the depth frame to the color information from the color frame so that the software is able to associate RGB color to each point in the depth frame and render an improved visualization in the Oculus Rift. The Kinect SDK provides both mapping functions and they are used accordingly in the software. The function to map depth frame to camera

space is declared as MapDepthPointToCameraSpace in the ICoordinateMapper struct. It receives a depth space point and a depth value and returns by reference the camera space point. The function to map the depth frame points to color frame points is declared as MapDepthPointToColorSpace. Similarly, it receives a depth space point and a depth value and returns by reference a color space point. With those two conversions, the software is able to place each point in a 3D space and associate RGB color to each point, creating the point cloud. Those function are presented in the frame rendering cycle, in Fig. 1

The next processing stage in Fig. 2 updates the point cloud data to their respective OpenGL buffers and to the Bullet Engine objects in order to create the point cloud graphic visualization and simulate interactions between the point cloud and the virtual environment. Figure 1 shows the order in which those structures and functions are activated in order to render a frame. This simulation is further discussed below.

The whole virtual environment is graphically displayed as a rectangular room. In this room, there is a green block and a colored ball within the user's field of action, i.e., objects with which the user's avatar can interact. However, this interaction can only happen through a physical contact which is simulated by the Bullet Engine. For this to happen, each object, such as the green block, the colored ball and even the walls of the room have to be represented as geometrical shapes inside the Bullet Engine, otherwise those elements would stay still or move in the room without ever noticing any contact with other objects, percolating through each other as if they did not exist. The Bullet Engine, as any other physics engine, has to predict the final rectangular and angular position of an object, given its initial position and velocity and using the geometrical representation of each object. This is how the physics simulation occurs and without it there could be no interaction whatsoever between the avatar and the virtual space and objects.

Assuming that user contact interactions happen mostly near joints like wrist, ankles, knees or elbows for example, it is sensible to consider that approximately any useful contact interaction happens through joints. The Kinect V2 provides 25 joints for tracking and all of them are used in the software implementation. In the Bullet Engine, each avatar joint (dealt with as if it were a point in camera space) becomes a small sphere, specifically a kinematic rigid body type (*Coumans, 2015*), capable of colliding with other objects, but not having its position altered by them. This representation is what allows the user to interact with other objects in the virtual room. Each joint has a position that is dictated solely by the user and, when each frame is being rendered, this joint position is given to their respective kinematic spheres. This implementation allows for real time tracking and modification of each joint's position in the physics simulation and leaves the task of calculating velocity, acceleration, momentum and other variables of the objects to the physics engine.

After every joint position is updated, the software advances one step with the simulation. If any object such as the green block or the sphere is moving or collides with other objects or with an avatar joint, the physics engine updates their position, rotation and other variables accordingly. This information is also sent to their respective OpenGL buffers, and now OpenGL is ready to render a new frame.

The final step is to get the motion data from the Oculus Rift as shown in Fig. 2. This action is needed to properly link the head motion with the OpenGL, so that there is a

correspondence for the HMD position and angles with what can be seen in the OpenGL virtual environment. In the end, the software sends the frame to the Oculus Rift. Figure 1 shows the roomScene struct calling the Render() function in order to finish the rendering cycle and display the final image in the Oculus Rift.

This method is novel in the way it uses several Kinect and Oculus Rift features integrated with graphical elements and physics simulations. As pointed out in the Introduction, there are few publications that tackle this integrated approach between HMDs and motion tracking devices (*Woodard & Sukittanon, 2015*; *Baldominos, Saez & Del Pozo, 2015*; *Stavroulia et al., 2016*), yet none of them developed a live avatar, with skeleton tracking and environment interactivity such as touching and pushing.

## Functionality and performance assessment

We conducted a comprehensive set of experiments to evaluate the functionality and performance of the method. First, we analyze the core aspects of avatar visualization with point cloud and polygon surface. The focus is on user experience and visualization perspectives. Rendering performance issues and limitations are assessed with the Fraps tool (Beepa Pty Ltd., Albert Park, Victoria, Canada), which provides the frame rate measurement. We also investigate the interactivity experience with respect to physical collisions of virtual objects with the avatar.

We then proceed with three tasks that are able to further evidence the properties of the system. For that, 2 participants were recruited: one who is experienced in interacting with virtual reality environments, and the other who is naive to usage of this technology. Both participants signed the informed consent forms (The Brazilian National Committee of Ethics in Research—CONEP- Ethics Committee protocol number: IRB 1.610.260).

In the first task, we evaluate whether the user avatar is capable of tracking the trajectory of a virtual object performing a 1-dimensional sinusoidal motion. Participants are seated in a comfortable chair and asked to wear the Oculus Rift HMD. After a fine adjustment of head and body position, the participant sees their arms (point cloud) and a small red virtual block. The block starts a 1-dimensional sinusoidal trajectory (0.015 rad/frame update), and the participant has to use their hand to follow the same trajectory as that from the block. Performance is evaluated by the coefficient of determination ($R^2$) between the user and the block trajectory.

In the second task, we evaluate whether the system provides a realistic interaction experience, i.e., if the actions performed by the participant are correctly captured by the Kinect sensor and displayed as a point cloud avatar that is able to interact with virtual 3D objects. To accomplish that, participants are asked to jiggle a green block from left to right for as long as they can, making sure that it does not fall. Performance is assessed by the number of successful jiggles.

In the third and last task, we investigate if our Oculus plus Kinect method performance is similar to that from other systems in terms of trajectory tracking and mismatches between the virtual and real world integration. In this task, the participant is positioned in front of a table, located between six pillars (Fig. 3A). The Kinect is placed at $\approx$1.5 m in front of the table, and six precision motion capture infrared cameras (OptiTrack s250e, NaturalPoint

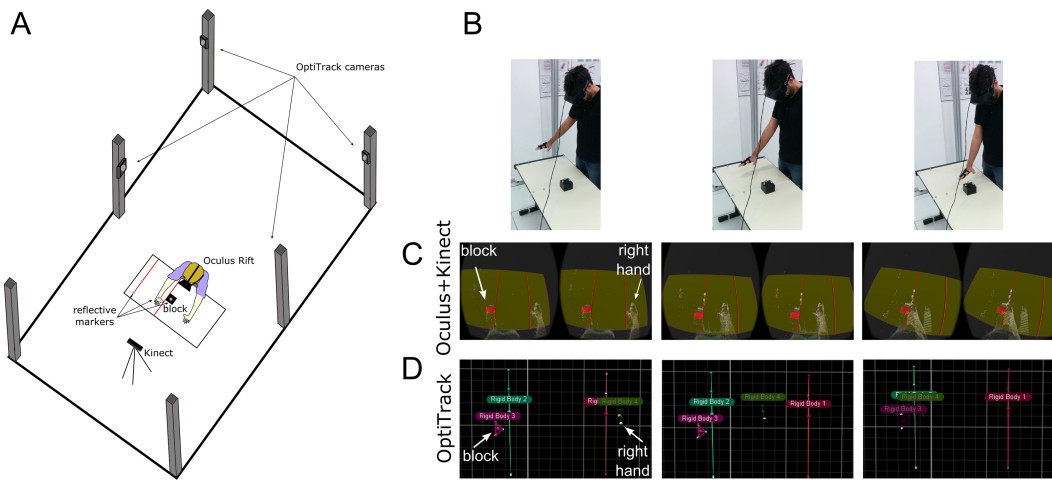

**Figure 3** Third task setup: (A) Block reaching task. In each trial, the participant places their right hand to the side of the rightmost line and then moves until reaching the black block. Six precision motion capture cameras are attached one to each pillar. The user wears the Oculus Rift HMD and the Kinect is positioned in front of the table. Three task stages (B) as captured by the Oculus plus Kinect (C) and OptiTrack (D) systems.

Inc.) are attached to the pillars (one to each pillar). On the table, we mark two lines (53 cm apart) and place a black block right to the side of the leftmost line. Then, one participant is asked to position their hand next to the rightmost line and move toward reaching the black block (Fig. 3B). The procedure is repeated for 120 s. Using reflective markers and the IR cameras, we are able to accurately measure the participant's hand and the block spatial coordinates, as well as detect when the hand crosses either of the lines. During the task, the participant is wearing the Oculus Rift and sees a corresponding virtual scenario (table and block), together with a point-cloud visualization of the right hand. In this way, we can compare the hand trajectory captured by the Oculus plus Kinect system with that captured by the OptiTrack system. Also, we can determine if there are any mismatches between the physical and real world by looking at the spatial coordinates over time of each task element.

## RESULTS

### A Method for Creating a user-resembling avatar

The software implementation of the described method results in a demonstration capable of creating and rendering a real-time avatar representation of the user. The implementation is written based on the samples provided by the Oculus Rift SDK, keeping the code elements that deal with the Oculus Rift communication and adding the code elements that communicate with the Kinect handler class, generate the avatar for the OpenGL library and execute the physics simulation for the Bullet Engine.

Using the pointcloud approach allows the software to implement two types of avatar: a pointcloud-based avatar and a polygon-based avatar. The first type is shown in Fig. 4. The software creates a pointcloud using the camera space points that the Kinect provides. Each camera space point becomes a point inside the OpenGL virtual environment with

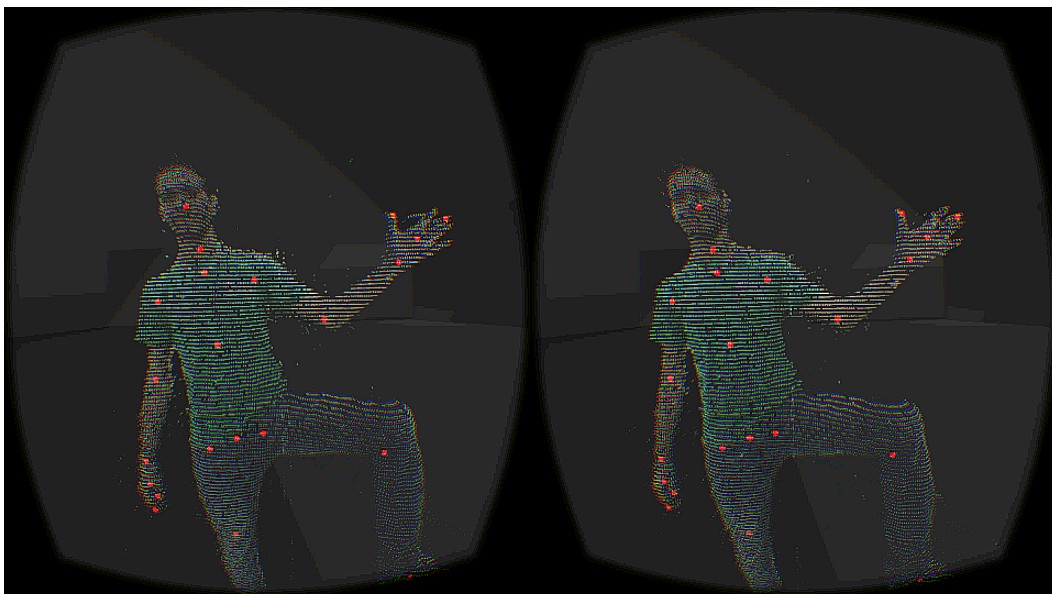

**Figure 4  Avatar visualization with point cloud.** This figure represents both left and right frames projected through the Oculus Rift lenses. The point-cloud avatar is seen in third-person perspective with a striped look, which is a consequence of the alignment of the point cloud and the position in which the frame was captured in 3D space. The red dots in the body indicate joint points provided by the Kinect and that are used for collisions.

properties of three-dimensional space and RGB color. The other type of avatar uses the camera points to create polygons where each vertex is a point in the three-dimensional space as show in Fig. 5. As a result, this implementation produces a sense of continuity in the avatar's body representation, only at the expense of knurled edges.

In addition, the software allows the user to see the virtual environment in first-person perspective. This feature is specially important for the immersion factor in our integrated hardware-software approach. The correct head location is obtained with the head position tracking provided by the Kinect SDK while the head rotation and angle is handled by the Oculus SDK. This coordination of features provides a solid recreation of the user's field of view while they walk around the physical environment. The only limitation is the Kinect field of view and the length of the Oculus Rift cable. In addition, the software allows for a fine adjustment on the head position with the arrow keys or the "a", "w", "d", "s" keys in the keyboard.

The Kinect also provides support for the recognition of up to six users and the software implements this feature. Every person in front of the Kinect sensor is recognized as an user (up to six) and the software offers the option of which user is on first-person mode. This choice is made with the number keys in the keyboard starting from 1 up to 6. The "0" key serves as a reset mode, detaching the field of view from the last chosen user.

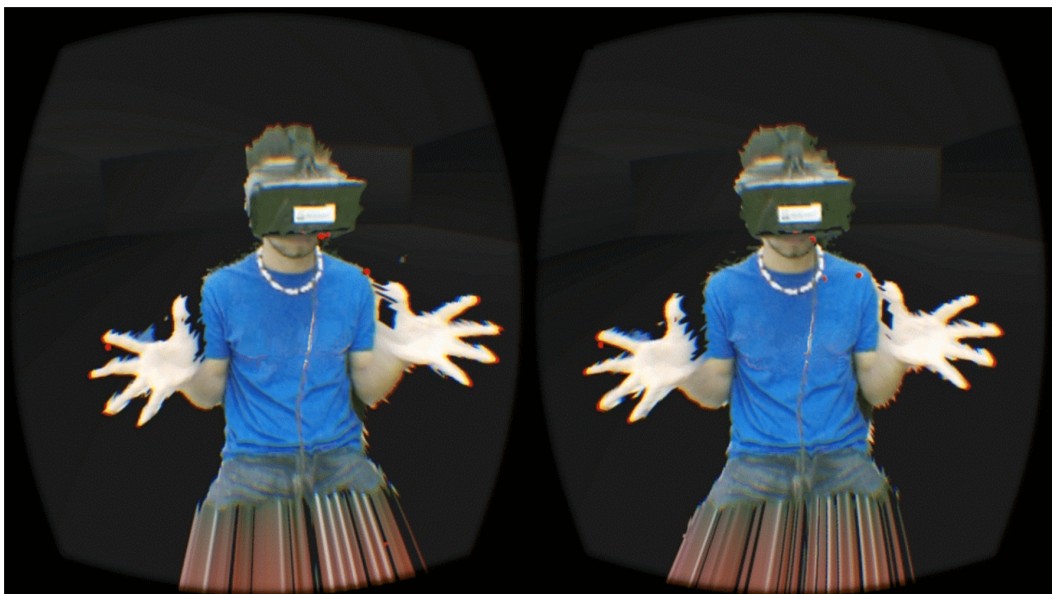

**Figure 5** **Avatar visualization with polygon surface.** Similarly to Fig. 4 we see both left and right frames projected through the Oculus Rift's lenses. In this case, the avatar has a continuous surface, except for its borders and generated artifacts on the base. The red dots that represent Kinect joints can still be noticed behind the avatar and its right index finger.

## Physical collisions with the Avatar

One of the keystones for an immersive virtual reality experience is interactivity. In this approach, we used the Bullet physics engine in order to recreate a desired physical interaction simulation between the avatar and an arbitrary virtual object.

A green solid rectangular block was created in the virtual environment and placed within the reach of the user's avatar. Figure 6 represents the moment when the user tries to balance the green block with their own hands, preventing it from falling. In the code, the programmer can add more objects to the simulation using the Bullet physics engine.

In our approach, each joint vertex provided by the Kinect becomes a bullet static object. It is possible to visualize a general distribution of joints along the body in Fig. 4, where each red dot is a joint. Therefore, each red dot can collide with virtual elements, considering the fact that the dot itself does not suffer any consequence from the collision because its location depends only on the data from the Kinect. In Fig. 6, the red dots are the contact points between the avatar's hands and the green block and are responsible for pushing and moving the object at any condition.

## Avatar interaction with the VR world

Avatar interaction with the VR world was assessed with three tasks. In the first task, the block tracking task, both naive and experienced users succeeded in following the sinusoidal block motion (Figs. 7A and 7C). Note the high resemblance between block and user trajectories in the $X$ direction ($R^2 = 0.99$), whilst the $Y$ and $Z$ directions present a more divergent motion.

We proceeded by investigating the user ability in handling a virtual object. Figures 7B and 7D show the results for the block jiggling task. Both naive and experienced users were

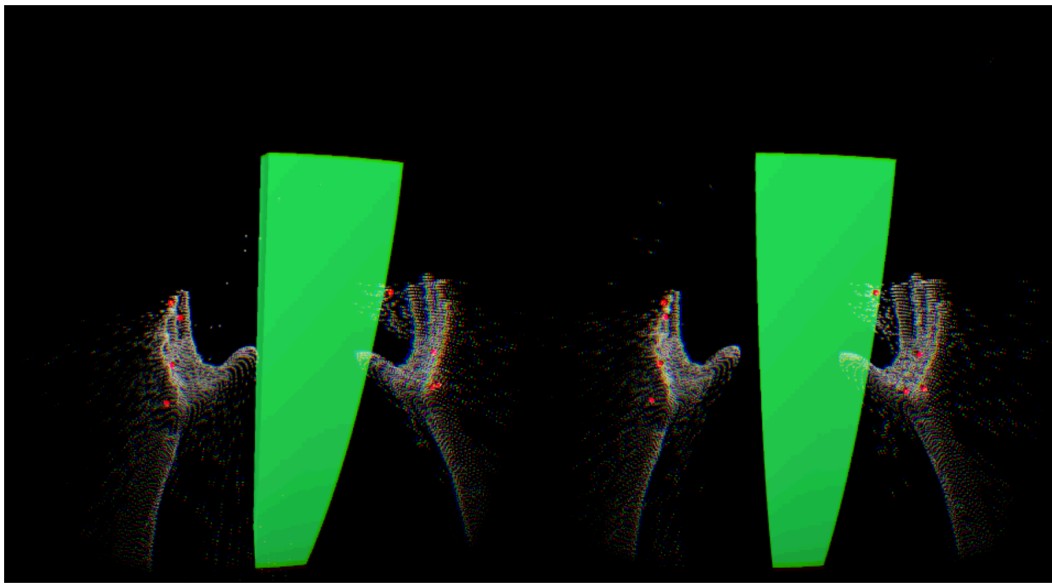

**Figure 6** **Physics interaction with green block.** This figure shows the first-person perspective while the avatar interacts with a green block. The avatar is reaching for the green block in order to interact with it. Although the hands are graphically displayed, only the red dots can contact the green block.

able to manipulate the object, but the latter was significantly better (95% confidence, as indicated by the non-overlapping boxplot notches) at preventing the block from falling.

Lastly, in the third task (Figs. 3 and 8), we investigated the tracking performance and analyzed possible mismatches between the real and VR worlds by comparing the Oculus plus Kinect system with a precision motion capture device (OptiTrack). As the participant moves their hand from one side of the table toward the black block, both systems successfully capture the trajectory, as displayed in Fig. 8A. Note that the OptiTrack trajectories are smoother than the Oculus plus Kinect trajectories, which is expected given the OptiTrack tracking method (infrared) and superior frame rate (120 frames/s). The moment both systems detect that the participant has reached the block has a difference of ≈30 ms (Fig. 8A, inset panel). Regarding the Oculus plus Kinect system frame rate over the course of the task, we observe from Fig. 8B that it has an average value of ≈44 frames/s; the minimum value was 35 frames/s. Note that the Kinect has a frame rate of 30 frames/s, thus there may be a mismatch between the Kinect and the Oculus data. Nevertheless, the results from the three tasks above indicate that this does not compromise the user experience.

Taken together, our results indicate that the method suggested in this paper provides a real time immersive experience interaction.

## DISCUSSION

### An approach for creating a real-time responsive and user-resembling avatar

This work presents a structured method for integrating both hardware and software elements in order to conceive an avatar visualization that is responsive to user movements

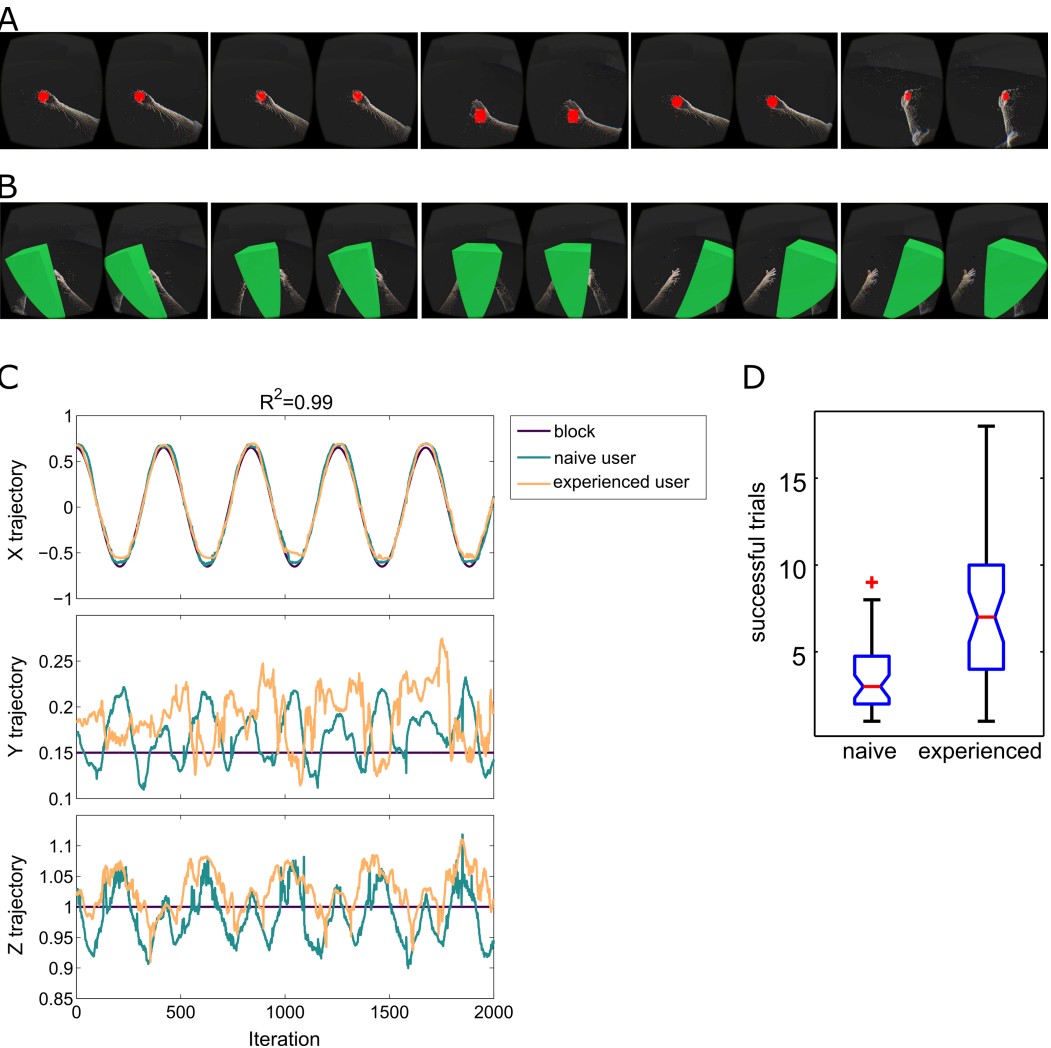

**Figure 7** **Avatar interaction with virtual world.** (A) Block tracking task. With their hand, the user has to follow a red block as it performs a 1D sinusoidal trajectory (left to right panels). (B) Block jiggling task. Using both hands, the user has to jiggle a parallelepiped-shaped green block from left to right whilst ensuring that it does not fall. (C) Representative 2,000 iterations of block and user trajectories for $X-Y-Z$ coordinates. Note that the block sinusoidal trajectory occurs only in the $X$ coordinate. (D) Distribution of successful trials (number of successive left to right block jiggles with no falls in $n = 43$ attempts) for a naive and experienced user.

and can interact with virtual objects. The implementation includes features such as first and third person perspective, the choice of which user is the main point of view, and also the possibility to adjust the point of view, as in farther or closer to the central head position, for example. The method was developed in order to increase the immersion experience in VR systems, focusing on avatar resemblance with the user and the interaction with the virtual environment.

Considering the three levels of self representation presented by *De França & Soares (2015)*, the system exploit mostly the body and identity levels. On one hand, the creation of the avatar assumes a body position and shape that is intrinsically connected with the

A

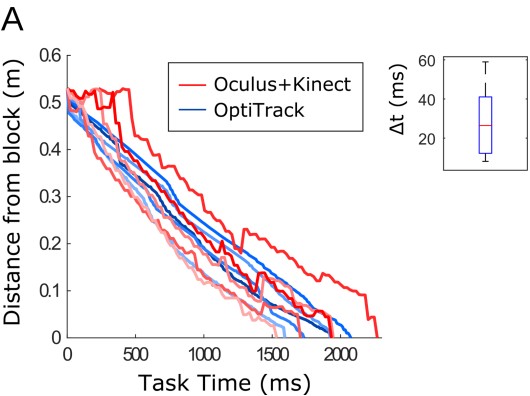

B

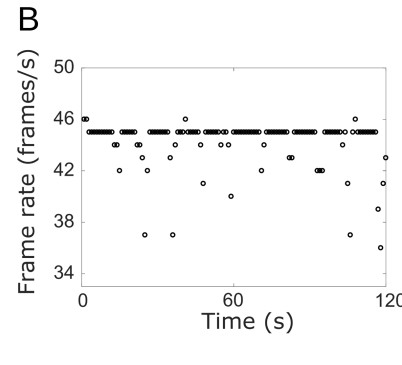

**Figure 8** **Comparison between the Oculus plus Kinect and OptiTrack systems.** (A) Five representative trials of hand trajectories (represented as hand distance from block over time) for each system. Each trial has a duration of 1.5–2 s. Blue and red color scale (lighter to darker) relate to respective trial number. The inset panel shows the time difference ($\Delta t$) between both systems in detecting that the block has been reached. (B) Oculus plus Kinect system frame rate over the duration of the task (120 s).

user to create a self representation. On the other hand, the identity level is supported by a feature that recreates the avatar from real time user data, allowing them to see their own shapes and clothes in real time. Also, the possibility to interact with objects allows the user to sense the limits between their virtual selves and the virtual environment, creating avatar identity. The last level of self-representation, affection, is not directly addressed by the system, as we do not shape or modify the avatar to increase affection. Yet it emerges from the affection the user has for their own virtual representation, since it resembles closely their real body shape. Therefore, the system presents a good combination of all the three levels, which impacts on immersion and sense of embodiment.

Exploring three different tasks, we were able to assess the system performance with respect to the immersion and interactivity requirements. In the first task (Fig. 7), the block tracking task, both naive and experienced users were able to follow the sinusoidal path performed by the block. This indicates that the avatar is easily incorporated by the user and its interaction with elements from the VR world are coherently coupled with actions in the real world. This fact is reinforced by the second task, in which users manipulate a virtual object. The Oculus plus Kinect system is capable of creating an avatar that is responsive to users actions whilst the collision detection module provides a realistic sense of interaction with the real world. The statistically significantly difference in performance between naive and experienced VR users in a relatively easy manual coordination task stresses that the interaction with virtual objects can improve with practice. Also, incorporating a haptic feedback system may improve user experience in the virtual world (*Lecuyer et al., 2004*; *Otaduy, Okamura & Subramanian, 2016*). Finally, in the third task (Fig. 8), results indicate that the mismatch between the virtual world and real world actions is kept to a minimum and does not compromise the user immersion experience.

## Limitations and further improvements
### Point cloud visualization and program parallelization
The core concept in this implementation involves the visualization of a point cloud. This visualization can be either as points floating in the virtual space for each data point in the point cloud or a surface recreation from the point cloud data. As a result, the quality of the avatar graphical visualization is inherently dependent on the point cloud data quality.

The point cloud data provided by the Kinect is noisy, as seen in Fig. 6. The implementation for the avatar visualization, in this work, sends all kinect point data to the OpenGL library, without performing any analysis or filtering. The result is an avatar visualization that is easily recognizable as the current user, but presents noisy and undefined borders. A first improvement over this method would be to implement a data filter to exclude points from the visualization that goes beyond a predefined body contour. A first method shown in *Point Cloud Library (2011)* involves trimming points that are located further than a specified threshold, the interval between the mean and the standard deviation of the global distance, according to a Gaussian distribution of the distance. Other methods (*Schall, Belyaev & Seidel, 2005*) involve heavy calculations and, although they might produce a better result, the filtered point cloud may not significantly improve visualization.

The point cloud itself imposes another problem that affects the speed of the system. During the rendering cycle, after the data is acquired from the Kinect, the system has to convert all the data points from the depth space to the camera space (i.e., the real world 3D dimensions), this processing is made in the CPU because it uses Kinect specific functions and this limits the processing throughput according to the number of cores the CPU has.

In addition, there are several methods to create a surface from the point cloud. The method used to create Fig. 5 consists of connecting the camera space points in groups of three, considering the 2D matrix from which they were acquired, and rendering a triangle with OpenGL. However, using the polygon rendering mode with OpenMP may cause the program to suffer a severe drop on rendered frames per second, thus further work should explore this issue. When creating a smooth surface from the point cloud, it is important to observe how the method will affect frames throughput on OpenGL engine. An optimal implementation would seek to parallelize this process within the graphics processing unit (GPU). The implementation of a point cloud triangulation algorithm such as presented in *Scheidegger, Fleishman & Silva (2005)* would have to be tested for real time applications.

### Integration with eye and gaze tracking technologies
One of the key factors for immersion, as mentioned by *Alshaer, Regenbrecht & O'Hare (2017)*, is the Field of View (FOV). In general, FOV or visual field is the area of space within which an individual can detect the presence of visual stimulus (*Dagnelie, 2011*). For VR, one might assume that matching the FOV of VR headsets with that of our own eyes would be ideal, but that is not necessarily the case.

A widening of the field of view is beneficial to the immersion factor as stated by *Prothero & Hoffman (1995)* and *Alshaer, Regenbrecht & O'Hare (2017)*. Nowadays, most VR headsets have a field of view of about 110 vertical degrees (*DigitalTrends, 2017*), whereas the field of

 

view of the human eye with binocular view is about 135 vertical degrees. However, there are other factors such Depth-of-Field (DoF) Blur that affect the perception of field of view. *Hillaire et al. (2008)* mention that technologies such as eye and gaze tracking can be used to improve the Depth-of-Field Blur, considering that DoF simulates the fact that humans perceive sharp objects only within some range of distance around the focal distance, and thus humans do not use the whole FOV at the same time. *Hillaire et al. (2008)* concluded that eye and gaze tracking technologies could be used to exploit visual effects that depend on the user's focal point, aiming to increase usage comfort and immersion within VR systems.

Nowadays, there are a few options that implement eye tracking capabilities in HMDs. The first one is Tobii Eye Tracking, which tries to perform custom adaptations to link an existing eye tracking device to a HMD. The second and most promising option would be the FOVE VR Headset, which implements the integration of HMD and eye tracking in the system as a whole. However, FOVE is currently open for pre-orders only and software development for this device is still in its initial steps.

### Alternative approach for an improved physical simulation

In its current version, each detected skeleton joint from the Kinect becomes a small sphere for physical simulation within the Bullet Engine. As a result, only a few predetermined points are able to interact with the virtual objects in the simulated environment. This implementation renders a realistic physical simulation when the user pushes a virtual object, but performance deteriorates when the user tries to hold objects, specially the small ones. This happens because the Kinect provides only three skeleton joints for the hand, which is not enough for a realistic approximation for hand interaction in some situations. This limitation is extended for the whole body for different interactions, when the number of points does not allow for a realistic approximation for body interactions.

An improvement over this implementation would be to use a greater number of points for approximating body interactions. In this approach, a much greater number of points from the point cloud would become spherical rigid bodies in the Bullet engine instead of limited skeleton joints. In this way, there would be a better sense of interaction with the body surface because the gap between contact points would decrease and improve the perception of continuity. Using a greater number of points is a reasonable alternative given that the Bullet Engine does not officially allow for kinematic surfaces in its current implementation, which discourages developing physical interactions with an actual surface obtained from the point cloud. However, as the number of rigid bodies in the simulation increases the Bullet Simulation slows down, which would affect the number of frames the application could provide for the Oculus Rift. A possible solution for this problem is in experimental stage and will be available in future versions of the Bullet Engine. The Bullet Engine developers are using OpenCL to parallelize simulation in the GPU and, as a result, the engine can handle real time simulation of thousands of objects simultaneously (*Harada, 2011*).

This alternative solution for physical interaction could improve the level of realism the software provides, specially the finesse with which each touch is calculated. The user would be able to handle objects with their fingers, for example, depending on how many points

from the point cloud are used, and whole-body interactions would be better illustrated for the body surface, instead of the body joint approximation.

## Interfacing with other platforms

In the last few years, virtual reality has become much more accessible. The Oculus Rift and the Kinect were surely promoters of this revolution by taking really expensive hardware used for scientific research and reshaping it to the consumer price level. Yet this revolution continues within the realm of mobile platforms, spreading even more the virtual reality concept to the masses. The Google Cardboard is a good example. For a reduced price, one can buy or build a head mounted display that uses their own smartphone as a display for a virtual reality environment.

In order to use the present system in a mobile setting, we have to consider how the system is organized and how the interface with each device works. If it were to substitute the Oculus Rift for a smartphone-based HDM, there is still the need to use the Kinect to create the avatar representation. However, building an interface to connect the Kinect directly with a mobile platform is challenging. A possible solution to this problem is to implement a webservice whose main task would be to pipe pre-processed Kinect frame data to the smartphone. In this architecture, the bulk of processing, which is converting depth points to 3D space, would be done in the desktop computer and a portable graphics rendering solution such as OpenGL ES would be able to use this data to create the virtual environment similarly to what happens in the Oculus Rift. A similar project was developed by *Peek (2017)*, in which he can send depth and skeleton data to a Windows Phone.

## CONCLUSION

VR systems are constantly improving, yet they still lack a few features that could enhance user experience. The immersion factor is crucial for user engagement, allowing for life-like psychological and physical user responses and scientific experiments.

There are still a few elements that influence the immersion experience. The avatar is one of them and, aligned with *Wrzesien et al. (2015)*, who showed that appearance resemblance with the user creates a stronger set of emotional responses, avatars that recreate user features will probably provide an improved virtual reality immersion.

This work aimed to create a user avatar with currently available commercial technology. We used Kinect and Oculus Rift for this purpose. The former is responsible for acquiring color, depth and movement data from the environment, which allows the latter to visually recreate an avatar representation of the user. Together with a physics engine, we provide avatar interactions with objects in the virtual environment.

The method presented in this work lies within the realm of VR applications, specially those which require avatar representation and physical interactions with the virtual environment. The hardware and software approach we used is expandable to other systems. It is innovative in the way it binds the hardware and software capabilities by integrating the virtual environment with skeleton tracking, body detection, and object collisions in order to create a unique, user-resembling and interactive avatar. All of these features together appeal to the final user and contribute to an immersive interaction with virtual reality.

## ACKNOWLEDGEMENTS

We acknowledge the support from the staff and students from the Edmond e Lily Safra International Institute of Neuroscience.

### Funding

This work was supported by the Brazilian Financing Agency for Studies and Projects (FINEP), the Brazilian Ministry of Science, Technology and Innovation (MCTI), the National Institute of Science and Technology (INCT) Brain Machine-Interface (INCEMAQ/MCTI/CNPq/CAPES/FAPERN), and the Ministry of Education (MEC). The funders had no role in study design, data collection and analysis, decision to publish, or preparation of the manuscript.

### Grant Disclosures

The following grant information was disclosed by the authors:
Brazilian Financing Agency for Studies and Projects (FINEP).
Brazilian Ministry of Science, Technology and Innovation (MCTI).
National Institute of Science and Technology (INCT).
Brain Machine-Interface (INCEMAQ/MCTI/CNPq/CAPES/FAPERN).
Ministry of Education (MEC).

### Competing Interests

The authors declare there are no competing interests.

### Author Contributions

- Igor Macedo Silva conceived and designed the experiments, performed the experiments, analyzed the data, wrote the paper, prepared figures and/or tables, performed the computation work, reviewed drafts of the paper.
- Renan C. Moioli conceived and designed the experiments, performed the experiments, analyzed the data, contributed reagents/materials/analysis tools, wrote the paper, prepared figures and/or tables, reviewed drafts of the paper.

### Ethics

The following information was supplied relating to ethical approvals (i.e., approving body and any reference numbers):

The Brazilian National Committee of Ethics in Research (Comissão Nacional de Ética em Pesquisa - CONEP) granted Ethical approval for the Santos Dumont Institute to carry out the study within its facilities (Ethical Application Ref: 1.610.260).

### Data Availability

GitHub: https://github.com/igormacedo/liveinteractiveavatar.

## Supplemental Information

Supplemental information for this article can be found online at http://dx.doi.org/10.7717/peerj-cs.128#supplemental-information.

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
