# Peer review of "A method for creating interactive, user-resembling avatars"

_PeerJ Computer Science, doi:10.7717/peerj-cs.128_

## Round 0.1 · original submission · Major Revisions

The reviews indicate several reservations from the reviewers, especially on the originality of the work. I would suggest the authors make a major revision to address all of the reviews comments and improve the justifications. A comparison with other state of the art method can improve experimental analysis and justification.

Reviewer 1 ·

Basic reporting

Given article proposes the method for addressing the problem of creating realistic graphical representation of the user for VR environment with available commercial technologies. In order to keep the review coherent, the comments for each sections are given following the structure of the paper.

Introduction is well-organized, coherent. In the first paragraphs definition of VR is introduced and its development history is shortly described. The key issue of control over immersion and interactivity is introduced and the possible solution using multiple technologies of motion tracking and HMDs is proposed.
[Line 27, 37, 38, 40] The abbreviation of head mounted devices spelled wrong
[Line 47]: Instead of “ideal system” it is suggested to say “Ideally, this system ..”
[Line 49-50] Stating the motivation incoherently, move the sentence to following paragraph to line 57 “ In accordance with growing evidence… ,… this article presents…”

In the “Background and related work’ section authors described technology of implementation, immersion factors and importance of user resembling avatar embodiment. It is suggested to include comparison of available VR development technology and choice justification.
[Line 75-79] Reference the technical specifications
[Line 84] Maybe “improved the first iteration” instead of “improves on the first iteration”?
[Line 108-109] how do those levels influence or was exploited in the presented system?

“Materials and methods” section describes software implementation and hardware use of the proposed method. It would be easier to follow the implementation process if authors would use dataflow (fig. 2) diagram during the explanation of core functionality of the system.
[Line 132-134] Repetition the information from background
[Line 155] Typo: “of” instead “on”
[Line 160]The concept of point cloud should be referenced
[Line 230-231]Reference the publications

In the “Results” section two types of avatar visualization and attempts of its interactivity implementation are presented. First and third perspectives of avatar are shown in figures.

Discussion’s subsections of avatar resemblance as an immersion factor and possible applications of VR could be moved to the introduction and background sections, leaving the limitation and further improvements for discussion. In this section it is also suggested to mention eye and gaze tracking technologies for improvement immersion quality as it influences FOV (line 100).
[Line 279-280] The features were not mentioned in the table of requirements introduced on line 151.

Experimental design

no comments

Validity of the findings

no comments

Additional comments

no comments

Reviewer 2 ·

Basic reporting

no comment

Experimental design

no comments

Validity of the findings

no comments

Additional comments

Well-known depth cameras and libraries to deal with complex physics interaction. The authors fail to properly describe what was the research outcome and impact. Novelty part is very weak.

Reviewer 3 ·

Basic reporting

The paper is written well, however, there are several grammatical issues throughout the text. The literature review is very relavant and follow well with the current state if the art.

The main issue that I see is that the contribution of the work is limited on a theoretical level. All the methods mentioned is the paper is already published. On an implementation side of things, the library provided is indeed useful for those working on developing VR tools.

Another main issue in the limitations imposed by point could data. The sensitivity of these can have a several practical implementation such a system on the quality and speed.

Experimental design

The experimental design is rather not very complete. While the approach adopted is to report the implementation details, it does not sufficiently address the performance issues and limitations.

The impact of reduced data points and the mismatchs needs be analyzed in detail. The simulations also needs to take into account different platforms such as that in mobile devices, which is growing popular for these applications.

The performance analysis also needs to include a large number of images and have a statistical analysis to be scientifically relavant. On the implementation side, some aspect of software testing and code optimization would also benefit to validate the usefulness of the method.

Validity of the findings

The novelty is not sufficiently justified. The new combination of methods seems to be the main scientific contribution, however, it is not enough contribution to justify publication. I would suggest the authors to justify and analyze the combination methods, and compare with other methods to justify the originality.

---

## Round 0.2 · accepted · Accept

The paper is Accepted, subject to further language polishing which can be performed while in production..

Reviewer 1 ·

Basic reporting

Authors addressed all comments given in the previous review. However, it is recommended to use language editing service to avoid stylistic issues throughout the paper.

Experimental design

no comment

Validity of the findings

no comment

Additional comments

no comment